# Ferroptosis, a Regulated Form of Cell Death, as a Target for the Development of Novel Drugs Preventing Ischemia/Reperfusion of Cardiac Injury, Cardiomyopathy and Stress-Induced Cardiac Injury

**DOI:** 10.3390/ijms25020897

**Published:** 2024-01-11

**Authors:** Vyacheslav V. Ryabov, Leonid N. Maslov, Evgeniy V. Vyshlov, Alexander V. Mukhomedzyanov, Mikhail Kilin, Svetlana V. Gusakova, Alexandra E. Gombozhapova, Oleg O. Panteleev

**Affiliations:** 1Laboratory of Experimental Cardiology, Department of Emergency Cardiology, Cardiology Research Institute, Tomsk National Research Medical Center, Russian Academy of Sciences, Tomsk 634012, Russia; rvvt@cardio-tomsk.ru (V.V.R.); evv@cardio-tomsk.ru (E.V.V.); sasha_m91@mail.ru (A.V.M.); kilin112233@gmail.com (M.K.); gombozhapova@gmail.com (A.E.G.); panteleev.o.o@yandex.ru (O.O.P.); 2Department of Biophysics and Functional Diagnostics, Siberian State Medical University, Tomsk 634050, Russia; gusacova@yandex.ru

**Keywords:** ferroptosis, heart, ischemia/reperfusion, kinases, cardiomyopathy, microRNAs

## Abstract

The hospital mortality in patients with ST-segment elevation myocardial infarction (STEMI) is about 6% and has not decreased in recent years. The leading cause of death of these patients is ischemia/reperfusion (I/R) cardiac injury. It is quite obvious that there is an urgent need to create new drugs for the treatment of STEMI based on knowledge about the pathogenesis of I/R cardiac injury, in particular, based on knowledge about the molecular mechanism of ferroptosis. In this study, it was demonstrated that ferroptosis is involved in the development of I/R cardiac injury, antitumor drug-induced cardiomyopathy, diabetic cardiomyopathy, septic cardiomyopathy, and inflammation. There is indirect evidence that ferroptosis participates in stress-induced cardiac injury. The activation of AMPK, PKC, ERK1/2, PI3K, and Akt prevents myocardial ferroptosis. The inhibition of HO-1 alleviates myocardial ferroptosis. The roles of GSK-3β and NOS in the regulation of ferroptosis require further study. The stimulation of Nrf2, STAT3 prevents ferroptosis. The activation of TLR4 and NF-κB promotes ferroptosis of cardiomyocytes. MiR-450b-5p and miR-210-3p can increase the tolerance of cardiomyocytes to hypoxia/reoxygenation through the inhibition of ferroptosis. Circ_0091761 RNA, miR-214-3p, miR-199a-5p, miR-208a/b, miR-375-3p, miR-26b-5p and miR-15a-5p can aggravate myocardial ferroptosis.

## 1. Introduction

The hospital mortality rate in patients with ST-segment elevation myocardial infarction (STEMI) is 4.6–7.5% and has not decreased in recent years [1,2,3,4,5]. Mortality is particularly high among patients with cardiogenic shock and microvascular obstruction [6,7]. It is quite obvious that there is an urgent need to create novel and more effective drugs for the treatment of acute myocardial infarction (AMI). The study of the molecular mechanism of ferroptosis could contribute to the creation of such drugs.

Until 1972, it was generally accepted that the only form of cell death is necrosis. However, after the discovery of apoptosis by Currie’s group [8], the state of affairs changed, and researchers got used to the idea that cell death could be regulated. If cell death is regulated, it can therefore be prevented or, on the contrary, stimulated. In 1972, Vladimirov and Archakov discovered a form of cell death that was distinguished by lipid peroxidation and the involvement of Fe^2+^ [9]. This form of cell death has been considered to play an important role in the pathogenesis of ischemic and reperfusion cardiac injury [10,11]. It has been assumed that lipid peroxidation induces the rupture of the cell membrane and cell death [10,11]. It has been suggested that antioxidants can prevent lipid peroxidation and cell death [10].

In 2012, Dixon et al. called this form of cell death “ferroptosis” [12]. They gave the following definition: “Ferroptosis is an iron-dependent form of nonapoptotic cell death” [12]. Dixon et al. suggested that ferroptosis is dependent upon intracellular Fe^2+^, triggered by erastin and inhibited by ferrostatin-1 [12]. Ferroptosis is a type of cell death triggered by oxidative stress [12]. Ferroptosis ends with plasma membrane rupture; in this respect, it is similar to necrosis.

## 2. The Main Manifestation of Ferroptosis

Ferroptosis ends with cell membrane rupture and intracellular protein release [10,11,12,13]. However, intracellular protein release occurs in necrosis, necroptosis, and pyroptosis [14,15,16]. Lipid peroxidation is accompanied by malondialdehyde (MDA), reactive oxygen species (ROS), and 4-hydroxynonenal (4-HNE) formation [10,11,17]. In one study, the MDA level was increased by 30–300% [17,18,19,20,21,22]. The 4-HNE level was increased by 40% [18]. Glutathione (GSH) was reduced by 70–97% in ferroptosis [18,19,23]. Some protein expression has also shown to be altered in ferroptosis. In other studies, the expression of acyl-CoA synthetase long-chain family member 4 (ACSL4) was increased by 130–300% [19,21,23]. Elsewhere, the expression of prostaglandin endoperoxide synthase 2 (PTGS2) was increased by 200–300% [21,24]. In some studies, the expression of the transferrin receptor (TFR-1) was reduced by 30% [18] or increased 2-fold [19,25]. The expression of cystine/glutamate transporter (SLC7A11) (xCT) was reduced by 90% [18]. In other studies, ferritin heavy chain-1 (FTH1) expression was decreased by 70–170% [19,24]. Elsewhere, the glutathione peroxidase-4 (GPX4) level was decreased by 40–70% [19,20,22,23,24,25]. However, some investigators could not find alterations in the GPX4 level in ferroptosis [18]. None of the listed markers are specific for ferroptosis. Therefore, investigators usually evaluate the four or five marker levels [17,19,21,23,24,25,26]. An important indicator of the involvement of ferroptosis in a pathological process is a decrease in its intensity after the use of deferoxamine, ferrostatin-1, UAMC-3203, dexrazoxane, and liproxstatin-1 which are ferroptosis inhibitors [12,27,28].

## 3. Inhibitors of Ferroptosis

The first articles in which the Fe^2+^ chelator deferoxamine was considered an inhibitor of ferroptosis were published 10 years ago [29,30]. However, the first articles which considered deferoxamine as an inhibitor of lipid peroxidation were published over 50 years ago [31,32]. In 2012, the first article that demonstrated the ability of ferrostatin-1 to inhibit ferroptosis was published [12]. In 2014, the first article that demonstrated the ability of liproxstatin-1 to inhibit ferroptosis was published [33]. The ability of UAMC-3203 to inhibit ferroptosis was later demonstrated by Devisscher et al. [34]. Ferrostatin-1 stability in rat plasma (% recovery after 6 h) is 1.1% [34]. UAMC-3203 stability in rat plasma (% recovery after 6 h) is 100% [34]. Therefore, UAMC-3203 is more effective in a long-term study than ferrostatin-1. In 2009, it was reported that the Fe^2+^ chelator dexrazoxane can mitigate anthracycline cardiotoxicity [35]. The investigators suggested that dexrazoxane prevents cardiomyocyte death, which is triggered by “ROS and iron”, after the application of anthracycline. Later, dexrazoxane was considered a pharmacological tool for studying ferroptosis [36]. It was reported that dexrazoxane can cross cell membranes and reduce the intracellular free Fe^2+^ level [28]. At present, deferoxamine, ferrostatin-1, UAMC-3203, liproxstatin-1, and dexrazoxane are used in the study of ferroptosis.

## 4. The Role of Kinases in the Regulation of Ferroptosis

It has been demonstrated that the activation of AMP-activated protein kinase (AMPK), extracellular signal-regulated kinase 1/2 (ERK1/2), phosphoinositide 3-kinases (PI3K), Akt kinase, protein kinase C (PKC), NO synthase (NOS), heme oxygenase-1 (HO-1), cyclooxygenase-2 (COX-2), and Janus kinase-2 (JAK2) promote an increase in cardiac tolerance to ischemia/reperfusion (I/R) [37]. In contrast, the stimulation of c-Jun N-terminal kinases (JNKs) and glycogen synthase kinase-3β (GSK-3β) contributes to a decrease in cardiac tolerance to I/R [37]. It could be hypothesized that these enzymes regulate the ferroptosis of cardiomyocytes. 

In one study, mice were subjected to CAO (60 min) and reperfusion (24 h) [38]. According to Gomez et al. (2008), an intravenous administration of the GSK-3β inhibitor SB216763 prior to reperfusion contributed to a decrease in infarct size by about 34% [38]. Ischemic postconditioning exhibited the same infarct-reducing effect [38]. The investigators argue that the infarct-reducing effect of postconditioning is a consequence of the phosphorylation (inactivation) of GSK-3β [38].

AMPK. It was found, in another study, that ferulic acid reduced infarct size in rat with coronary artery occlusion (CAO, 30 min) and reperfusion (120 min) [39]. Ferulic acid simultaneously inhibited ferroptosis in myocardial tissue [39]. Pretreatment with compound C, an AMPK inhibitor, abolished the inhibition of ferroptosis and the infarct size reduction in rats [39]. Glycation end-products stimulated ferroptosis in isolated rat cardiomyocytes [40]. Ferrostatin-1 and deferoxamine inhibited ferroptosis. This effect was eliminated by compound C [40]. Consequently, AMPK is involved in the cytoprotective effect of ferroptosis inhibitors. One study reported that Puerarin, an active ingredient in the traditional Chinese medicine Pueraria, inhibited lipopolysaccharide (LPS) and induced myocardial ferroptosis in rats [19]. This effect was abolished by compound C [19]. Consequently, AMPK is involved in inhibition of ferroptosis. Another study found that ischemia/reperfusion induced ferroptosis in an isolated rat heart [41]. The α2-adrenergic receptor (α2-AR) agonist, dexmedetomidine, suppressed ferroptosis. This effect was abolished by compound C [41]. According to one study, canagliflozin, a sodium–glucose cotransporter-2 inhibitor, alleviated palmitic acid-induced ferroptosis of the HL-1 cardiomyocyte cell line [42]. Compound C eliminated this effect of canagliflozin. Another study reported that embryonic rat heart-derived H9c2 cells were exposed to H_2_O_2_ which induced ferroptosis of these cells [43]. Idebenone, an analog of coenzyme Q10 (CoQ10), mitigated ferroptosis. The investigators obtained evidence that AMPK could be involved in the inhibition of ferroptosis [43]. In another study, it was found that CAO (30 min) and reperfusion (24 h) induced ferroptosis in a rat heart [44]. Britanin, a bioactive sesquiterpene lactone isolated from *Inula lineariifolia*, reduced infarct size, alleviated ferroptosis, and increased the p-AMPK level in myocardial tissue [44]. These data demonstrated the involvement of AMPK in inhibition of ferroptosis (Figure 1).

ERK1/2. In one study, it was reported that hypoxia/reoxygenation (H/R) induced ferroptosis of H9c2 cells [45]. Dexmedetomidine alleviated ferroptosis, increased cell viability, and triggered the phosphorylation (activation) of ERK1/2. The inhibition of ERK1/2 by U0126 reversed the cytoprotective effect of dexmedetomidine and mitigated the dexmedetomidine-triggered suppression of ferroptosis [45]. Consequently, ERK1/2 is involved in the inhibition of ferroptosis (Figure 1). 

Protein kinase A (PKA). As we have already reported above, H/R induced ferroptosis of H9c2 cells [45]. Dexmedetomidine partially reversed this effect. The PKA inhibitor H89 eliminated the inhibition of ferroptosis by dexmedetomidine. SiRNA against CREB also partially reversed the dexmedetomidine-triggered inhibition of ferroptosis, where CREB is a cAMP response element-binding protein. The investigators concluded that dexmedetomidine alleviated H/R injury of H9c2 cells by suppressing ferroptosis through the activation of the cAMP/PKA/CREB signaling pathway (Figure 1).

PKC. One study found that doxorubicin and erastin, a ferroptosis inducer, resulted in ferroptosis of H9c2 cells [46]. Pretreatment with the E-prostanoid 1 receptor agonist 17-PT-PGE2 increased cell viability and inhibited ferroptosis [46]. The PKA and PKC inhibitor staurosporine (20 nM/L) reversed the 17-PT-PGE2-triggered inhibition of ferroptosis [46]. It should be noted that staurosporine, at a final concentration of 20 nM/L, completely blocks PKC and partially inhibits PKA [47,48]. Consequently, it could be argued that the activation of PKC promotes the inhibition of ferroptosis (Figure 1).

Akt. In one study, H9c2 cells were subjected to H/R [49]. H9c2 cells were transfected with a microRNA miR-199a-5p inhibitor to down-regulate miR-199a-5p and an miR-199a-5p mimic to up-regulate miR-199a-5p prior to H/R. The miR-199a-5p inhibitor increased cell viability, suppressed ferroptosis, and increased the p-Akt/Akt ratio [49]. The Akt inhibitor LY294002 abolished the cytoprotective effect of the miR-199a-5p inhibitor and reduced the p-Akt/Akt ratio [49]. Consequently, the activation of Akt promotes the inhibition of ferroptosis and an increase in cell viability in H/R. In another study, it was found that doxorubicin induced a cardiotoxic effect which is accompanied by ferroptosis [22]. LCZ696, an angiotensin receptor and neprilysin inhibitor, protects the rat heart against doxorubicin and suppresses ferroptosis. The Akt inhibitor LY294002 alleviates the cardioprotective and anti-ferroptotic effects of LCZ696. Investigators have suggested that sirtuin-3, a soluble mitochondrial NAD-dependent deacetylase, is involved in the cardioprotective effect of LCZ696. It has been shown that LCZ696 protects H9c2 cells against the cytotoxic effect of doxorubicin and inhibits doxorubicin-induced ferroptosis. Sirtuin-3 knockout abolishes both protective effects of LCZ696. In addition, these investigators found that LCZ696 stimulates the expression of superoxide dismutase-2 (SOD2). They concluded that the cardioprotective effect of LCZ696 is mediated via the activation of the Akt/sirtuin-3/SOD2 pathway [22]. Thus, the stimulation of Akt alleviates ferroptosis of cardiomyocytes (Figure 1).

NOS. We have already reported above that the miR-199a-5p inhibitor suppressed ferroptosis and increased H9c2 cell survival in H/R through the activation of Akt [49]. It was found that the miR-199a-5p inhibitor increased the concentration of NO in a culture supernatant of H9c2 cells [49]. The miR-199a-5p inhibitor increased the p-eNOS/eNOS ratio. The Akt inhibitor LY294002 abolished an increase in the p-eNOS level. The investigators concluded that the miR-199a-5p inhibitor increased H9c2 cell tolerance to H/R through the stimulation of the Akt/eNOS pathway. However, the anti-ferroptotic effect of canagliflozin is accompanied by a decrease in the inducible NOS (iNOS) mRNA level in HL-1 cells [42]. Thus, there is no definition of the role of NOS in the regulation of ferroptosis in myocardial tissue (Figure 1).

PI3K. One study reported that doxorubicin induced the death and ferroptosis of H9c2 cells [23]. Pretreatment with 740Y-P, a PI3K activator, mitigated both effects of doxorubicin and increased HO-1 expression [23]. Lapatinib, an ErbB-2 and EGFR tyrosine kinase inhibitor, enhanced doxorubicin-induced ferroptosis and reduced the p-Akt level in H9c2 cells [23]. In another study, it was found that trastuzumab, an anticancer drug, induced ferroptosis of cardiomyocytes both in vivo and in vitro and also reduced the p-PI3K/PI3K ratio [50]. Ferrostatin-1 and deferoxamine inhibited ferroptosis of cardiomyocytes and increased the p-PI3K/PI3K ratio [50]. It has also been reported that suberosin exhibits cardioprotective and anti-ferroptotic effects which are associated with an increase in the PI3K mRNA in rats pretreated with the ferroptosis inducer thiazolidinedione [51]. Suberosin is a natural product that is isolated from the roots and aerial parts of *Cudrania tricuspidata* [51]. These data demonstrate that the stimulation of PI3K promotes the inhibition of ferroptosis (Figure 1). 

COX-2. Zhang et al. (2023) did not find convincing evidence of the involvement of COX-2 in the regulation of palmitic acid-induced ferroptosis in HL-1 cells [42].

HO-1. Doxorubicin-induced ferroptosis is associated with an increase in the HO-1 mRNA level in murine hearts [36]. Sepsis-induced ferroptosis is accompanied by an increase in HO-1 expression in the murine heart [52]. The cardioprotective effect of the α2-AR agonist dexmedetomidine in mice with sepsis has been associated with a decrease in HO-1 expression in the murine myocardium [52]. One study reported that sickle cell disease induced cardiomyopathy and ferroptosis in the murine heart and promoted the upregulation of HO-1 in myocardial tissue [53]. The inhibition of HO-1 by tin protoporphyrin-IX caused the suppression of ferroptosis in mice with SCD. In contrast, the induction of ferroptosis promoted HO-1 expression in mice [53]. In another study, it was found that the chronic administration of di(2-ethylhexyl) phthalate (DEHP) induced ferroptosis in the murine heart [54]. This effect is associated with an increase in HO-1 expression in myocardial tissue. One study reported that doxorubicin induced ferroptosis and increased HO-1 expression in HL-1 cells [55]. HMOX1 knockdown vector (HMOX1 short hairpin RNA (shRNA)) reduced HO-1 expression and inhibited the ferroptosis of HL-1 cells [55]. Cardiac-specific Sirtuin 1 knockout aggravated the cardiotoxic effect of doxorubicin and ferroptosis in mice [55]. Both effects were accompanied by an increase in HO-1 expression in myocardial tissue [55]. It was reported elsewhere that the HO-1 inhibitor zinc protoporphyrin suppressed isoproterenol-induced myocardial ferroptosis [56]. It was found, in another study, that the cytoprotective and anti-ferroptotic effects of the MiR-432-5p mimic are associated with an increase in the HO-1 level in isolated cardiomyocytes subjected to H/R [57]. 

These data demonstrate that HO-1 is involved in the pathogenesis of ferroptosis of cardiomyocytes (Figure 1).

GSK-3β. It was shown, in one study, that britanin reduced infarct size, alleviated ferroptosis, and increased the p-GSK-3β level in myocardial tissue of rats with I/R of the heart [58]. Bian et al. (2023) showed that palmitic acid induced ferroptosis of human cardiomyocyte AC16 cells [59]. This effect was associated with a reduction in the phosphorylation of Akt and GSK-3β. Celastrol, a bioactive compound isolated from the herb *Tripterygium wilfordii*, inhibited ferroptosis and increased cell viability. Celastrol simultaneously triggered the phosphorylation of Akt and GSK-3β. The investigators suggested the Akt/GSK-3β signaling pathway participated in the anti-ferroptotic and cytoprotective effects of celastrol [59]. According to Gomez et al. (2008), the phosphorylation-induced inactivation of GSK-3β plays a negative role in cardiac tolerance to reperfusion [38]. Consequently, a decrease in the phosphorylation (activation) of GSK-3β could promote ferroptosis. Bian et al. (2023) [59] did not provide an explanation for this discrepancy and did not discuss Gomez’s data. Consequently, the role of GSK-3β in the regulation of ferroptosis requires further study.

In summary, these data demonstrate that the activation of AMPK, HO-1, ERK1/2, PKA, PKC, Akt, and PI3K promotes the inhibition of ferroptosis. In contrast, the simulation of GSK-3β contributes to the ferroptosis of cardiomyocytes.

## 5. The Role of Non-Coding RNA in the Regulation of Ferroptosis in the Heart

In recent years, much attention has been paid to studying the role of non-coding RNAs (ncRNAs), particularly microRNAs, long non-coding RNAs (lncRNAs), and circular RNAs in the pathogenesis of cardiovascular diseases, in particular, in the regulation of ferroptosis [60,61]. 

It was found, in one study, that lncRNA Snhg7 plasmid induced ferroptosis of -1 cells via the activation of T-box transcription factor 5 (Tbx5) [62]. It was demonstrated, in another study, that the serum level of small extracellular vesicle-encapsulated (SEMA5A-IT1) RNAs negatively correlated with the serum creatine kinase-MB (CK-MB) level in patients with a cardiopulmonary bypass [63]. SEMA5A-IT1 RNAs are lncRNAs. Human cardiomyocyte AC16 cells were exposed to H/R. The cells were transfected by the lentiviral vectors of SEMA5A-IT1. These lncRNAs increased cell survival, inhibited apoptosis and ferroptosis through an increase in miR-143-3p expression. The miR-143-3p mimic exhibited the same cytoprotective effects as lncRNAs [63].

In one study, erastin, a ferroptosis inducer, induced ferroptosis and the death of H9c2 cells [64]. H9c2 cells were transfected with a lentiviral vector expressing miR-190a-5p which inhibits GLS2 gene expression (this gene encodes the synthesis of glutaminase 2). miR-190a-5p overexpression increased cell viability and inhibited ferroptosis. In contrast, anti-miR-190a-5p decreased cell survival and enhanced erastin-induced ferroptosis [64]. In a different study, mice underwent permanent CAO for 3 days [65]. CAO induced an increase in the miR-15a-5p level 2-fold. The investigators suggested that miR-15a-5p could regulate the tolerance of cardiomyocytes to H/R [65]. HL-1 cells were exposed to hypoxia for 24 h which induced the death of 30% of cells. MiR-15a-5p aggravated hypoxia-induced cell death through a reduction in GPX4 expression and an increase in the MDA and ROS levels in HL-1 cells. The investigators concluded that miR-15a-5p could be involved in the development of I/R cardiac injury through the activation of ferroptosis [65]. One study reported that Erastin induced ferroptosis and the death of HL-1 cells [66]. It was found that circRNA1615 reduced the cytotoxic effect of erastin. Investigators have proposed that the cytoprotective effect of circRNA1615 is a result of its anti-ferroptotic effect [64]. In a different study, rats underwent permanent CAO [67]. The duration of CAO was 28 days. An adverse remodeling of the heart was developed which was accompanied by the activation of ferroptosis. CAO induced an increase in miR-375-3p content in myocardial tissue by approximately 4-fold. It was found that miR-375-3p inhibited GPX4 expression. Ferrostatin-1 and the miR-375-3p inhibitor suppressed ferroptosis and improved the contractility of the heart. The investigators suggested that miR-375-3p induced ferroptosis through the inhibition of GPX4 expression, and the miR-375-3p inhibitor alleviated this process and prevented the adverse remodeling of the heart [67]. We have reported above that the miR-199a-5p inhibitor increased H9c2 cell viability and suppressed ferroptosis in H/R through the activation of Akt [49]. In one study, cultured rat cardiac microvascular endothelial cells (CMEC) were subjected to hypoxia [19]. Exosomes were isolated from the incubation medium of CMEC and added to H9c2 cells exposed to H/R. H/R induced ferroptosis and the death of H9c2 cells. The exosomes increased cell viability and inhibited ferroptosis. These exosomes contained miR-210-3p. The exosomes inhibited erastin-induced cell death and ferroptosis. The miR-210-3p inhibitor abolished the cytoprotective and anti-ferroptotic effects of exosomes. In another study, the miR-210-3p mimics suppressed ferroptosis [21]. Elsewhere, cultured human cardiac myocytes were subjected to hypoxia (1% O_2_) for 24 h [68]. Hypoxic cardiomyocytes secreted exosomes containing miR-208a/b. Erastin induced ferroptosis of cultured human cardiac fibroblasts (CFs). Exosomes enhanced erastin-induced ferroptosis of CFs. miR-208a/b inhibitors reversed the pro-ferroptotic effect of exosomes. The investigators concluded that hypoxic cardiomyocyte-derived exosomes can aggravate ferroptosis of CFs through miR-208a/b expression [68]. In a different study, exosomes isolated from the plasma of mice with permanent CAO inhibited erastin-induced ferroptosis and increased the survival of the Lewis lung carcinoma cell line LLC and osteosarcoma cell line K7M2 [69]. These exosomes contained miR-22-3p. This microRNA inhibited erastin-induced ferroptosis and increased tumor cell viability [69]. These data demonstrate that miR-22-3p is an inhibitor of ferroptosis. 

One study showed that hypoxia induced ferroptosis and the death of H9c2 cells [44]. It was found that miR-26b-5p mimics aggravated hypoxia-induced cell death and stimulated ferroptosis of H9c2 cells [44]. In a different study, the miR-214-3p level was increased in the infarcted region of the murine heart and in neonatal rat cardiomyocytes (NRCMs) subjected to hypoxia [70]. An increase in miR-214-3p content is accompanied by ferroptosis, and in this study, the miR-214-3p inhibitor (antagomir) improved cardiac contractility, reduced infarct size, and alleviated ferroptosis in myocardial tissues. Consequently, miR-214-3p induced ferroptosis of NRCMs. The miR-214-3p inhibitor protected NRCMs against hypoxia. The investigators suggested that malic enzyme 2 is a target of miR-214-3p. They proposed that miR-214-3p is an endogenous trigger of ferroptosis which suppresses malic enzyme-2 expression [70]. 

In one study, lipopolysaccharide from Escherichia coli induced sepsis-like cardiomyopathy in mice [71]. miR-130b-3p overexpression improved the contractility of the septic heart, reduced the serum creatine kinase-MB (CK-MB) and cardiac troponin I (cTnI) levels, and inhibited ferroptosis in myocardial tissue. LPS induced ferroptosis and the death of H9c2 cells. The miR-130b-3p mimic inhibited ferroptosis and increased cell viability [71]. In contrast, the miR-130b-3p inhibitor decreased cell viability and stimulated ferroptosis [71]. H/R caused an increase in the circ_0091761 RNA level in H9c2 cells. As found in one study, ferrostatin-1 resulted in a decrease in lactate dehydrogenase (LDH) release, decreased circ_0091761 expression, and inhibited ferroptosis of H9c2 cells [72]. The circ_0091761 inhibitor (si-circ_0091761) increased H9c2 cell viability and suppressed ferroptosis. H/R caused an increase in miR-335-3p content in H9c2 cells [72]. Si-circ_0091761 increased miR-335-3p expression in H9c2 cells in H/R. miR-335-3p mimics increased cell viability and inhibited ferroptosis. The investigators concluded that circ_0091761 enhanced H/R-induced cell death and ferroptosis and that circ_0091761 and an miR-335-3p mimic could protect the heart against I/R [72]. Elsewhere, HL-1 cells were exposed to hypoxia (1% O_2_) for 18 h [73]. Hypoxia reduced miR-450b-5p content in HL-1 cells and induced ferroptosis. miR-450b-5p mimics increased cell viability, reduced cTnI release from HL-1 cells, and suppressed ferroptosis in these cells [73].

In one study, neonatal rat ventricular cardiomyocytes were exposed to H/R [57]. H/R induced cell death and ferroptosis. An miR-432-5p mimic plasmid increased cell viability and inhibited ferroptosis [57]. The cytoprotective effect of the miR-432-5p mimic was associated with an increase in the expression of nuclear factor erythroid 2-related factor 2 (Nrf2). In addition, the miR-432-5p mimic increased HO-1 expression in cardiomyocytes and decreased it in the Kelch-like ECH-associated protein 1 (Keap1) protein level. It was reported that Keap1 is an endogenous inhibitor of Nrf2 [57]. It was found that miR-432-5p-Lipo reduced infarct size by about 30% and inhibited ferroptosis in myocardial tissue in rats with CAO (30 min) and reperfusion (4 h) [57]. The investigators concluded that the miR-432-5p mimic inhibits ferroptosis through the activation of Nrf2 and HO-1 expression in cardiomyocytes and the inhibition of Keap1 expression [57]. 

Thus, circ_0091761 RNA, lncRNA Snhg7, miR-214-3p, miR-199a-5p, miR-208a/b, miR-375-3p, miR-26b-5p and miR-15a-5p can aggravate ferroptosis. In contrast, miR-190a-5p, circRNA1615, miR-22-3p, miR-450b-5p, miR-130b-3p, miR-335-3p, miR-432-5p, miR-143-3p, SEMA5A-IT1 RNAs and miR-210-3p can inhibit ferroptosis. These data demonstrate that miR-450b-5p, miR-432-5p and miR-210-3p can increase the tolerance of cardiomyocytes to hypoxia/reoxygenation through the inhibition of ferroptosis. Circ_0091761 RNA, miR-214-3p, miR-199a-5p, miR-375-3p, miR-26b-5p, miR-335-3p, and miR-15a-5p can aggravate H/R-induced injury of cardiomyocytes through the enhancement of ferroptosis (Figure 2). 

## 6. The Role of Transcription Factors in the Regulation of Ferroptosis

Nuclear factor erythroid 2-related factor 2 (Nrf2). Nrf2 is a transcription factor that regulates the transcription of cell antioxidant defense genes; therefore, it could be suggested that Nrf2 can regulate and suppress ferroptosis [74].

In one study, it was found that H/R induced ferroptosis of H9c2 cells which was associated with a reduction in the Nrf2 mRNA level [75]. In another study, adriamycin resulted in cardiomyopathy which was accompanied by ferroptosis and increased NADPH-oxidase (NOX2, NOX4) expression in the myocardial tissue of rats [76]. Astragaloside IV reversed cardiomyopathy and ferroptosis and increased Nrf2 and GPX4 protein content in the myocardium. The investigators suggested that the anti-ferroptotic effect of astragaloside IV is mediated via the activation of the Nrf2/GPX4 signaling pathway [76]. One study found that the intravenous administration of antioxidant histochrome (1 mg/kg) prior to reperfusion reduced infarct size by about 30% and improved contractility of the heart in rats with CAO (60 min) and reperfusion (24 h) [77]. Histochrome inhibited ferroptosis in vivo. Histochrome increased the tolerance of isolated cardiomyocytes to oxidative stress induced by H_2_O_2_ and increased Nrf2 expression in these cells. It was proposed that the anti-ferroptotic effect of histochrome is mediated through an increase in Nrf2 expression and the activation of GPX4 [77]. In another study, it was reported that H/R caused ferroptosis of H9c2 cells [78]. Icariin, a flavonoid extracted from epimedii, increased cell viability and inhibited ferroptosis of H9c2 cells. Icariin inhibited erastin-induced ferroptosis and reduced the cytotoxic effect of erastin. H/R resulted in a reduction in the Nrf2 and HO-1 mRNA levels in H9c2 cells by about 70%. Icariin reversed this negative effect of H/R. The investigators suggested that the anti-ferroptotic effect of icariin is mediated via the stimulation of the Nrf2/HO-1 signaling pathway [78]. In another study, it was found that erastin caused ferroptotic H9c2 cell death [26]. Gossypol acetic acid (GAA), a natural product taken from the seeds of cotton plants, prevented ferroptosis and cell death. An isolated rat heart was subjected to global ischemia (30 min) and reperfusion (2 h). GAA increased GPX4 protein expression but reduced Nrf2 protein expression in myocardial tissue in I/R [26]. These data demonstrate that an increase in cardiac tolerance to ferroptosis could be developed without the involvement of Nrf2. In one study, rats underwent CAO (30 min) and reperfusion (4 h) [79]. H9c2 cells were exposed to H/R. Naringenin, a flavonoid from citrus fruits, reduced infarct size and inhibited ferroptosis in the rat heart. Pretreatment with naringenin increased the Nrf2 and GPX4 protein levels in myocardial tissue. Naringenin increased cell survival in H/R, inhibited ferroptosis, and increased Nrf2 and GPX4 protein content in H9c2 cells [79]. The investigators concluded that naringenin inhibits ferroptosis of cardiomyocytes through the activation of the Nrf2/GPX4 signaling pathway [79]. 

One study reported that doxorubicin induced the death and ferroptosis of H9c2 cells [80]. The cytotoxic effect of doxorubicin was associated with a decrease in the p62, Nrf2, GPX4, and HO-1 protein levels (p62 is a nuclear pore glycoprotein) [80]. Resveratrol alleviated the cytotoxic effect of doxorubicin, inhibited ferroptosis, and increased p62, Nrf2, GPX4, HO-1 protein content in H9c2 cells. Ferrostatin-1 also mitigated the cytotoxic effect of doxorubicin, inhibited ferroptosis, and increased GPX4 and HO-1 expression, but had no effect on the p62 and Nrf2 levels. The investigators proposed that the p62-Nrf2/GPX4/HO-1 signaling pathway is involved in the anti-ferroptotic effect of resveratrol [80]. GPX4 and HO-1 are involved in the anti-ferroptotic effect of ferrostatin-1 [80]. In a different study, it was found that xanthohumol, a flavonoid isolated from Humulus lupulus, protected H9c2 and neonatal cardiomyocytes against ferroptosis induced by Fe-SP ([N, N′-disalicylidene-1,2-phenylenediamine] Fe^3+^) [20]. The anti-ferroptotic effect of xanthohumol was accompanied by an increase in GPX4 expression and a decrease in Nrf2 expression in cardiomyocytes. An isolated rat heart was subjected to ischemia (30 min) and reperfusion. Xanthohumol reduced infarct size, inhibited ferroptosis, increased the GPX4 protein level, and reduced Nrf2 content in myocardial tissue [20]. These data demonstrate that an increase in Nrf2 expression is not a prerequisite for the inhibition of ferroptosis. One study reported that streptozotocin-induced diabetes resulted in ferroptosis in the myocardial tissue of mice [81]. Another found that the chronic administration of sulforaphane, an activator of Nrf2, alleviated diabetic cardiomyopathy in mice and inhibited ferroptosis [40]. AMPK knockout abolished the anti-ferroptotic effect of sulforaphane [40]. Consequently, AMPK is involved in the anti-ferroptotic effect of Nrf2 activation. It was reported that doxorubicin caused ferroptosis in the murine heart [82]. This effect was accompanied by the downregulation of Nrf2 expression [82].

In one study, doxorubicin was shown to induce the cardiomyopathy and ferroptosis of cardiomyocytes [83]. This effect was associated with a decrease in protein arginine methyltransferase-4 (PRMT4) expression and a reduction in isolated cardiomyocytes’ viability. Cardiomyocyte-specific PRMT4 overexpression was induced by an intravenous administration of adeno-associated virus 9 (AAV9) carrying PRMT4 under the cTnT promoter (AAV-PRMT4). Cardiomyocyte-specific PRMT4 knockdown was induced by an intravenous administration of short hairpin RNA (shRNA) against PRMT4 (AAV-shPRMT4) [83]. PRMT4 overexpression aggravated doxorubicin-induced cardiomyopathy and stimulated ferroptosis. In contrast, PRMT4 knockdown alleviated doxorubicin-induced cardiomyopathy and inhibited ferroptosis. PRMT4 overexpression reduced Nrf2 and GPX4 expression. PRMT4 knockdown increased Nrf2 and GPX4 expression [83]. These data demonstrate the important role of Nrf2 and GPX4 in the regulation of ferroptosis. We have reported above that dexmedetomidine inhibits ferroptosis [41]. The anti-ferroptotic effect of dexmedetomidine is associated with an increase in Nrf2 and GPX4 expression. In one study, rats underwent CAO (30 min) and reperfusion (120 min) [84]. Shenmai injection (SMJ), a traditional Chinese medicine, was injected intraperitoneally at the onset of reperfusion. Ferrostatin-1 and ML385, an inhibitor of Nrf2, were administered intraperitoneally 30 min before CAO. SMJ reduced IS and improved contractility in reperfusion, and ferrostatin-1 increased cardiac tolerance to I/R and inhibited ferroptosis. In contrast, erastin aggravated I/R cardiac injury and ferroptosis. SMJ increased the GPX4 and Nrf2 protein levels in myocardial tissue. Pretreatment with ML385 abolished the anti-ferroptotic effect of SMJ [84]. These results show the important role of GPX4 and Nrf2 in the regulation of ferroptosis. It was shown elsewhere that 6-gingerol, a polyphenol extracted from ginger, inhibited streptozotocin-induced ferroptosis in murine hearts and upregulated GPX4, Nrf2, and HO-1 expression [85] These data confirm the important role of GPX4, Nrf2, and possibly HO-1 in the inhibition of ferroptosis. In a different study, it was reported that 5-Fluorouracil, a potent antitumor agent, induced cardiomyopathy and ferroptosis in the myocardium of mice [86]. Ferrostatin-1 alleviated the cardiotoxic effect of 5-fluorouracil, inhibited ferroptosis, and increased GPX4 and Nrf2 expression [86].

In one study, sepsis was induced by cecal ligation, and a puncture caused septic cardiomyopathy in rats [87]. Sepsis was associated with the activation of cardiac ferroptosis. Ferrostatin-1 and resveratrol, a polyphenol compound extracted from strawberries, grapes, and peanuts, alleviated septic cardiomyopathy, inhibited ferroptosis, and increased GPX4 expression in the rat heart. Resveratrol increased the levels of Nerf2 and Sirtuin-1. The investigators suggested that the anti-ferroptotic effect is a result of the stimulation of the Sirt1/Nrf2 signaling pathway [87]. We have reported above that doxorubicin and erastin induced ferroptosis of H9c2 cells [46]. Pretreatment with 17-PT-PGE2 increased cell viability, inhibited ferroptosis, and increased GPX4 and Nrf2 protein expression [46]. Erastin reduced the GPX4 level in H9c2 cells. Furthermore, 17-PT-PGE2 reversed this effect of erastin and increased GPX4 content in H9c2 cells. The inhibitor Nrf2 ML385 abolished a 17-PT-PGE2-induced increase in the GPX4 level [46]. Consequently, it could be proposed that the anti-ferroptotic effect of 17-PT-PGE2 is mediated through the stimulation of the Nrf2/GPX4 signaling pathway.

Thus, the stimulation of the Nrf2/GPX4 and Nrf2/AMPK signaling pathways can prevent the development of ferroptosis. However, an increase in cardiac tolerance to ferroptosis could be developed without the involvement of Nrf2.

Signal transducer and activator of transcription 3 (STAT3) is a transcription factor that increases cardiac tolerance to I/R [37].

One study reported that doxorubicin resulted in the death and ferroptosis of HL-1 cells [88]. Ferroptosis was accompanied by a reduction in the p-STAT3/STAT3 ratio. The STAT3 inhibitor c188-9 aggravated doxorubicin-induced ferroptosis [88]. The STAT3 activator colivelin inhibited doxorubicin-induced ferroptosis [88]. Consequently, the stimulation of STAT3 inhibited myocardial ferroptosis.

Nuclear factor kappa-light-chain-enhancer of activated B cells (NF-κB) is a transcription factor which is involved in the cardioprotective effect of delayed ischemic preconditioning [37]. NF-κB is responsible for the regulation of genes involved in inflammation and immune responses [89]. 

One study found that lipopolysaccharide induced cardiac injury and myocardial ferroptosis in rats, increased interleukin-1β (IL-1β), IL-6, and tumor necrosis factor-1α (TNF-1α) levels in the heart and serum [24]. In addition, LPS caused an increase in the p-NF-κB-p65/NF-κB-p65 ratio and tall-like receptor-4 (TLR4) expression in the rat heart. Ferrostatin-1 alleviated LPS-induced cardiac injury, inhibited ferroptosis, reduced the pro-inflammatory cytokine levels, and decreased TLR4 and p-p65 expression in myocardial tissue [24]. In another study, heat shock (HS) of H9c2 cells was induced by the impact of high temperature (43 °C for 2 h) [90]. HS resulted in ferroptosis and an increase in TLR4, NF-κB, IL-1β, and IL-6 expression. Liproxstatin-1 reversed these effects of HS. TLR4 inhibitor TAK-242 or NF-κB inhibitor pyrrolidine dithiocarbamate alleviated HS-induced ferroptosis [90]. The investigators concluded that inhibition of the TLR4/NF-κB signaling pathway can alleviate ferroptosis of cardiomyocytes.

These data convincingly demonstrate that ferroptosis is associated with an increase in pro-inflammatory cytokine production and the stimulation of TLR4 and NF-κB expression. The activation of TLR4 and NF-κB promotes ferroptosis of cardiomyocytes.

## 7. Inflammation 

It was reported that sepsis-induced ferroptosis is accompanied by an increase in the serum and myocardial IL-6 levels in mice [52]. LPS induced cardiomyopathy and ferroptosis which are accompanied by an increase in the IL-6, IL-1β, and TNF-1α levels in the murine myocardium and serum [17,19,91]. Streptozotocin-induced cardiomyopathy is associated with the activation of myocardial ferroptosis, an increase in IL-6, IL-1β, and TNF-1α expression in the heart and an increase in serum IL-6, IL-1β and TNF-1α concentration in mice [85].

These data demonstrate that ferroptosis is associated with myocardial inflammation. Both processes are interconnected. However, it is unclear whether ferroptosis is a trigger for inflammation or whether inflammation is a trigger for ferroptosis.

## 8. Ischemia/Reperfusion of Cardiac Injury

In one study, it was found that retreatment with ferrostatin-1 reduced infarct size in mice with CAO (30 min) and reperfusion (24 h) [36]. In another study, an isolated murine heart was exposed to global ischemia (30 min) and reperfusion (2 h) [92]. I/R induced cardiac injury and led to a decrease in GPX4 expression. The ferroptosis inhibitor liproxstatin-1 reduced infarct size and increased the GPX4 level [92]. In a different study, rats underwent CAO (30 min) and reperfusion (2 h) [93]. Streptozotocin-induced diabetes aggravated I/R cardiac injury. Ferrostatin-1 reduced infarct size in rats [93]. CAO (30–60 min) and reperfusion (2–24 h) caused ferroptosis in myocardial tissue in rats and mice [39,79,84,94,95,96,97,98,99,100,101]. According to Tang et al. (2021), deferoxamine (200 mg/kg intraperitoneally) had no effect on infarct size and did not alter ferroptosis in myocardial tissue [94]. We also could not find an infarct-reducing effect of deferoxamine (60 mg/kg intravenously) in rats with CAO (45 min) and reperfusion (120 min) [102]. However, deferoxamine at a dose of 60 mg/kg abolishes the infarct-sparing effect of hypoxic preconditioning [102]. Consequently, it could be hypothesized that Fe^2+^ is not a ferroptosis rate-limiting factor. It is possible that even a small Fe^2+^ content in a cell is completely enough to induce ferroptosis in I/R of the heart.

It has been found that H/R triggers ferroptosis of H9c2 cells [25,78,103,104,105]. In one study, it was found that deferoxamine inhibited this process [103]. It was reported elsewhere that miR-15a-5p is involved in the development of I/R cardiac injury through the activation of ferroptosis [65]. In another study, it was shown that antioxidant histochrome inhibited ferroptosis and reduced infarct size in rats with CAO (60 min) and reperfusion (24 h) [77]. A different study reported that CAO (30 min) resulted in myocardial ferroptosis and contractile dysfunction in mice [66]. Ferrostatin-1 improved cardiac contractility and inhibited ferroptosis [66]. In one study, it was found that H/R caused ferroptosis of H9c2 cells [21,26,45,49,75,78]. It was also reported in one study that ferrostatin-1 increased cell viability and alleviated ferroptosis [49]. Another found that I/R of an isolated rat heart resulted in ferroptosis [106]. A different study reported that permanent CAO (28 days) induced myocardial ferroptosis which was alleviated by ferrostatin-1 [67]. In one study, isolated neonatal rat cardiomyocytes were exposed to hypoxia which caused the ferroptosis and oxidative stress of these cells [107]. Isolated rat hearts were subjected to ischemia (30–40 min) and reperfusion (60–120 min) [41,108]. This I/R triggered myocardial ferroptosis [41,108]. One study found that pretreatment with ferrostatin-1 alleviated I/R cardiac injury and inhibited ferroptosis in mice with CAO (30 min) and reperfusion (2 h) [94].

In one study, rats underwent CAO (2, 4, and 6 h) and reperfusion (3, 6, 12, and 24 h) [109]. 

It was found that ischemia contributed to an increase in the serum CK-MB level in proportion to the duration of ischemia. However, CAO had no effect on the MDA, F2+, GPX4, and FTH1 levels in myocardial tissue. The MDA, Fe^2+^, GPX4, and FTH1 levels were increased in reperfusion. MDA content reached its maximum after CAO (2 h) and 12 h of reperfusion and decreased 24 h after the restoration of coronary perfusion. GPX4 content reduced after CAO (2 h) and 6 h of reperfusion. The FTH1 level reached its maximum after CAO (2 h) and 6 h of reperfusion [109]. Pretreatment with ferrostatin-1 (3 mg/kg, intraperitoneally) before reperfusion reduced infarct size and prevented an increase in the serum CK-MB level [109]. These data demonstrated that reoxygenation triggered the process of ferroptosis that is involved in reperfusion cardiac injury.

These data convincingly show that I/R induces the activation of myocardial ferroptosis. Ferrostatin-1 and liproxstatin-1 inhibit ferroptosis and increase cardiac tolerance to I/R. Consequently, ferroptosis is involved in I/R cardiac injury. Deferoxamine has no effect on infarct size. It is possible that intracellular Fe^2+^ content does not limit the ferroptosis rate in I/R of the heart, and even a small intracellular Fe^2+^ concentration is enough to trigger ferroptosis in reperfusion of the heart.

## 9. Chemotherapeutic Agent-Induced Cardiomyopathy

Many antitumor drugs induce cardiomyopathy in cancer patients [110]. It is an important problem for the treatment of cancer. There is evidence that ferroptosis is involved in the development of this doxorubicin-induced cardiomyopathy and the toxic injury of isolated cardiomyocytes, H9c2 cells, and HL-1 cells [22,23,36,46,55,80,82,83,88,111,112,113,114]. Indeed, the ferroptosis inhibitor ferrostatin-1 prevents the appearance of doxorubicin-induced cardiomyopathy and ferroptosis and increases cell viability [36,46,80,83,88,113]. Ferroptosis participates in adriamycin-induced cardiomyopathy [76]. 5-Fluorouracil induces cardiomyopathy and ferroptosis in the myocardium of mice [86]. Ferrostatin-1 alleviates its cardiotoxic effect [110]. Ferroptosis is involved in the cardiotoxic effect of trastuzumab, an anticancer drug [50]. The Fe^2+^ chelator dexazoxane mitigates doxorubicin-induced cardiomyopathy and ferroptosis [115]. It was reported that deferoxamine (250 mg/kg) mitigates doxorubicin-induced cardiomyopathy and suppresses ferroptosis [116].

Thus, these data demonstrate convincing evidence that ferroptosis participates in the development of chemotherapeutic agent-induced cardiomyopathy.

## 10. Septic Cardiomyopathy

There is evidence that ferroptosis is involved in sepsis-induced cardiomyopathy [117]. Cecal ligation and puncture induces septic cardiomyopathy and ferroptosis in the murine myocardium [52,87]. Ferrostatin-1 alleviates septic cardiomyopathy and inhibits ferroptosis [87]. Lipopolysaccharide from Escherichia coli causes sepsis-like cardiomyopathy and ferroptosis in mice [17,19,24,71,81,118]. Ferrostatin-1 alleviates LPS-induced cardiomyopathy and inhibits ferroptosis [24,71,87,118]. The Fe^2+^ chelator dexrazoxane also mitigates LPS-induced cardiomyopathy and suppresses ferroptosis [118]. LPS causes cell death and the ferroptosis of H9c2 cells and neonatal rat cardiomyocytes [119].

These data demonstrate that ferroptosis could be involved in septic cardiomyopathy.

## 11. Diabetic Cardiomyopathy 

There is evidence that ferroptosis participates in diabetes-induced cardiomyopathy [120,121]. Streptozotocin induces diabetic cardiomyopathy and ferroptosis in the myocardial tissue of mice [40,75,81,85,122,123]. The ferroptosis inhibitor liproxstatin-1 prevents the development of diabetic cardiomyopathy [40]. Ferrostatin-1 and deferoxamine alleviates cardiomyopathy and ferroptosis [40]. The combination of a high-fat diet and low-dose streptozotocin induces cardiomyopathy [124]. Deferoxamine alleviates streptozotocin-induced cardiomyopathy [124]. A high-fat diet (HFD) induces a metabolic syndrome (MS)-like state with cardiac anomalies and ferroptosis in rats [125]. An MS-like state (type 2 diabetes) develops in db/db mice and is accompanied by cardiomyopathy and ferroptosis [126,127]. Ferrostatin-1 inhibits cardiomyopathy and ferroptosis [126]. HFD causes an MS-like state (type 2 diabetes) which is associated with cardiomyopathy and ferroptosis in mice [59,128]. 

In summary, diabetes mellitus causes cardiomyopathy and ferroptosis in myocardial tissue. It is suggested that ferroptosis is involved in the pathogenesis of diabetic cardiomyopathy.

## 12. Stress-Induced Cardiac Injury

Takotsubo syndrome (TTS) is distinguished by contractile dysfunction and usually affects the apex of the heart without coronary artery obstruction. TTS is distinguished by an increase in the blood levels of myocardial necrosis markers, microvascular dysfunction, and myocardial edema [129]. Stress-induced cardiomyopathy (TTS) is a rare disease. It is observed in 0.6–2.5% of patients with acute coronary syndrome [129]. However, the hospital mortality among patients with TTS corresponds to 3.5–12%, which is equivalent to the mortality of patients with STEMI [129]. The incidence of takotsubo syndrome is nine times higher in women aged 60–70 years than in men. Not all, but 70–80% of patients with TTS had physical or emotional stress that preceded this disease. Most patients with TTS have neurological or psychiatric illnesses. In patients with TTS, catecholamine levels are elevated, so it is believed that the occurrence of TTS is associated with excessive activation of the adrenergic system and contractile dysfunction. Stress-induced cardiac injury (SICI) is the result of the activation of the β1-adrenergic receptor (β1-AR) by endogenous catecholamines in rats [130].

There is indirect evidence that ferroptosis is involved in SICI. The β1- and β2-AR agonist isoproterenol induces the death and ferroptosis of H9c2 cells [131]. Ferrostatin-1 increases H9c2 cell viability. The administration of isoproterenol at a dose of 5 mg/kg subcutaneously for 14 days induces cardiomyopathy and myocardial ferroptosis [131]. Isoproterenol (100 µM) causes the death and ferroptosis of isolated neonatal rat cardiomyocytes [56]. Ferrostatin-1, liproxstatin-1, and deferoxamine increase cell tolerance to the cytotoxic effect of isoproterenol. Antioxidant N-acetylcysteine inhibites an isoproterenol-induced decrease in the GPX4 level in cardiomyocytes. The administration of isoproterenol (50 mg/kg/day subcutaneously) for 3 weeks results in cardiomyopathy and cardiac fibrosis which is accompanied by ferroptosis [56]. Ferrostatin-1 (1 mg/kg/day) for 3 weeks reduces the serum cTnI level and prevents the development of isoproterenol-induced contractile dysfunction, cardiac fibrosis, and ferroptosis. Isoproterenol increases the HO-1 level in isolated cardiomyocytes and in myocardial tissue. The administration of the HO-1 inhibitor zinc protoporphyrin (5 mg/kg/day) for 3 weeks reduces the serum cTnI level and prevents the development of isoproterenol-induced contractile dysfunction, cardiac fibrosis, and ferroptosis [56]. 

It has been reported that SICI is associated with an increase in the myocardial conjugated diene and MDA levels in rats [132,133,134]. Antioxidant butylated hydroxytoluene (ionol) mitigates SICI in rats [132]. The β1- and β2-AR antagonist propranolol abolishes lipid peroxidation [132]. The investigators did not detect other markers of ferroptosis. Therefore, it could be proposed, but not claimed, that stress causes ferroptosis.

These data demonstrate that the chronic activation of β1- and β2-ARs promotes the development of myocardial ferroptosis which triggers cardiomyopathy. HO-1 is involved in isoproterenol-induced cardiomyopathy and ferroptosis. Stress induces lipid peroxidation in myocardial tissue which is abolished by propranolol and ionol. These data indirectly demonstrate that ferroptosis could be involved in SICI.

## 13. Unresolved Issues and Prospects for the Use of Ferroptosis Inhibitors for the Treatment of Cardiovascular Diseases

The following constitute the therapeutic landscape of the use of anti-ferroptotic compounds for the treatment of cardiovascular diseases: deferoxamine, ferrostatin-1, liproxstatin-1. There is no convincing evidence of the cardioprotective effect of deferoxamine in I/R of the heart. Therefore, performing a clinical trial of the use of deferoxamine for the treatment of AMI is inappropriate. Ferrostatin-1 and liproxstatin-1 have the greatest promise for clinical use. Ferrostatin-1 increases cardiac tolerance to I/R. Ferrostatin-1 mitigates doxorubicin-induced cardiomyopathy. However, it is unclear whether ferrostatin-1 can aggravate cancer progression. Ferrostatin-1 alleviates septic cardiomyopathy and diabetic cardiomyopathy. Ferrostatin-1 increases cell resistance to the cytotoxic effect of isoproterenol. These data suggest that ferrostatin-1 could protect the heart against stress-induced injury. Liproxstatin-1 also augments cardiac tolerance to I/R. It mitigates diabetic cardiomyopathy. However, its cardioprotective effect in sepsis or doxorubicin-induced cardiomyopathy has not been evaluated. Liproxstatin-1 increases cardiac resistance to the cardiotoxic effect of isoproterenol. These findings suggest that liproxstatin-1 could increase cardiac tolerance to stress. Thus, there is convincing evidence of a need for clinical trials of ferrostatin-1 and liproxstatin-1 for the treatment of AMI, takotsubo syndrome, sepsis, and diabetes. A comparative analysis of ferrostatin-1 and liproxstatin-1 efficacies for the treatment of experimental cardiovascular pathologies has not been performed. Therefore, it is unclear which of the two compounds is more effective. The main disadvantage of both liproxstatin-1 and ferrostatin-1 is their poor solubility in water. Therefore, it is impossible to use these compounds for intravenous administration in humans in acute pathologies such as AMI, takotsubo syndrome, and sepsis. It is necessary to create water-soluble ferroptosis inhibitors similar to liproxstatin-1 and ferrostatin-1.

Calcium overload plays an important role in reperfusion cardiac injury [15]. However, the involvement of an Ca2+ overload in the pathogenesis of ferroptosis has not been studied before. There is only indirect evidence for the involvement of ferroptosis in the pathogenesis of SICI. Studies using ferroptosis inhibitors are needed.

## 14. Conclusions

The stimulation of PKA, AMPK, ERK1/2, PKC, PI3K, and Akt promotes the inhibition of ferroptosis. In contrast, the activation of HO-1 contributes to the development of myocardial ferroptosis (Figure 3 and Figure 4). The role of GSK-3β and NOS in the regulation of ferroptosis requires further study.

Circ_0091761 RNA, lncRNA Snhg7, miR-214-3p, miR-199a-5p, miR-208a/b, miR-375-3p, miR-26b-5p and miR-15a-5p can aggravate myocardial ferroptosis. In contrast, miR-190a-5p, circRNA1615, miR-22-3p, miR-450b-5p, miR-130b-3p, miR-335-3p, miR-432-5p, miR-143-3p, SEMA5A-IT1 RNAs and miR-210-3p can inhibit ferroptosis. miR-450b-5p and miR-210-3p can increase the tolerance of cardiomyocytes to hypoxia/reoxygenation through the inhibition of ferroptosis (Figure 3 and Figure 4). 

The activation of the Akt/sirtuin-3/SOD2, cAMP/PKA/CREB, Nrf2/GPX4, and Nrf2/AMPK signaling pathways can prevent the development of ferroptosis (Figure 3 and Figure 4). In some cases, cardiac tolerance to ferroptosis could be developed without the involvement of Nrf2. The stimulation of STAT3 inhibits myocardial ferroptosis. Ferroptosis is associated with an increase in pro-inflammatory cytokine production and the stimulation of TLR4 and NF-κB expression in the heart. The activation of TLR4 and NF-κB promotes ferroptosis of cardiomyocytes. Ferroptosis is associated with myocardial inflammation. However, it is unclear whether ferroptosis is a trigger for inflammation or whether inflammation is a trigger for ferroptosis.

It has been convincingly shown that I/R induces the activation of myocardial ferroptosis. The ferroptosis inhibitors increase cardiac tolerance to I/R. Consequently, ferroptosis is involved in I/R cardiac injury. Deferoxamine does not alter infarct size. It is possible that intracellular Fe^2+^ content does not limit the ferroptosis rate in I/R of the myocardium. Ferroptosis participates in the development of chemotherapeutic agent-induced cardiomyopathy. Ferroptosis could be involved in septic cardiomyopathy. Ferroptosis is involved in the pathogenesis of diabetic cardiomyopathy. The chronic activation of β1- and β2-ARs promotes the development of myocardial ferroptosis and cardiomyopathy. HO-1 participates in isoproterenol-induced cardiomyopathy and ferroptosis. There is indirect evidence that ferroptosis could be involved in SICI (Figure 5).

## Figures and Tables

**Figure 1 ijms-25-00897-f001:**
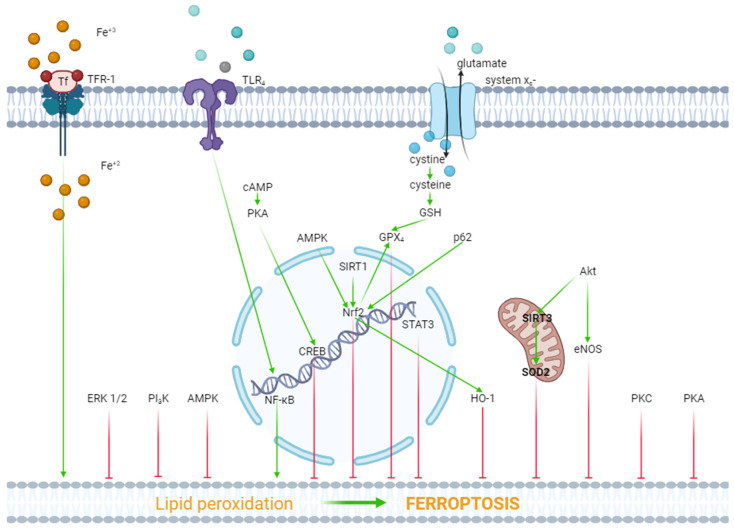
The role of kinases and transcription factors in the regulation of ferroptosis. Red arrows are inhibitors of ferroptosis, and green arrows are inducers of ferroptosis.

**Figure 2 ijms-25-00897-f002:**
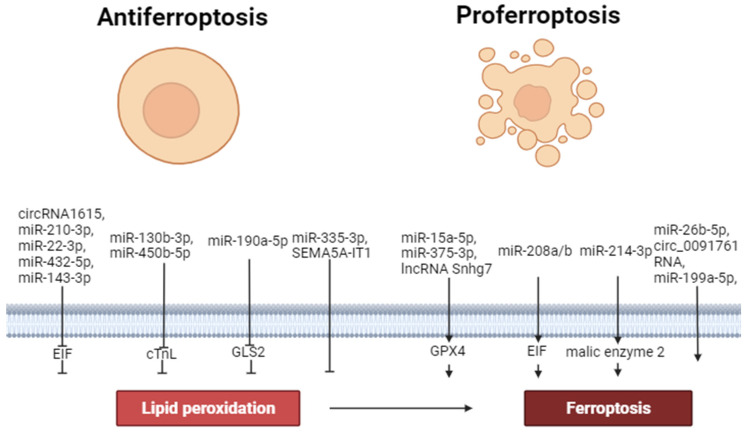
The role of non-coding RNAs in the regulation of ferroptosis. Perpendicular arrows are inhibitors of ferroptosis, arrows are inducers of ferroptosis.

**Figure 3 ijms-25-00897-f003:**
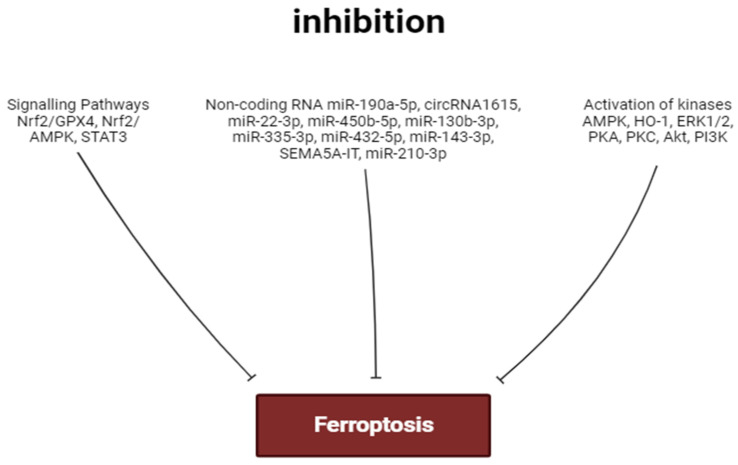
Factors inhibiting ferroptosis. Perpendicular arrows are inhibition of ferroptosis.

**Figure 4 ijms-25-00897-f004:**
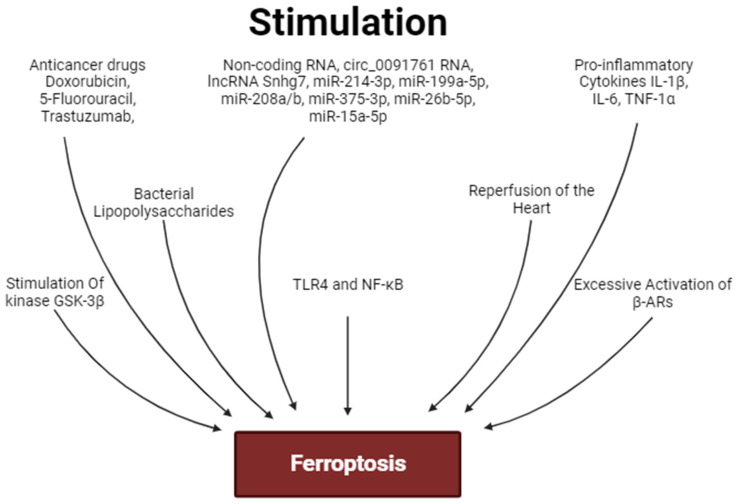
Factors stimulating ferroptosis. Arrows are stimulation of ferroptosis.

**Figure 5 ijms-25-00897-f005:**
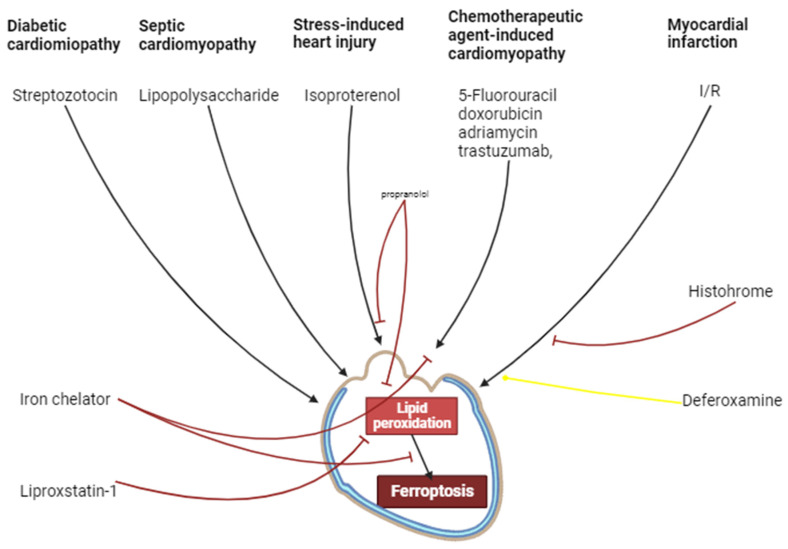
The involvement of ferroptosis in cardiovascular pathologies. Black arrows are stimulation of processes, red and yellow arrows are inhibition of processes.

## Data Availability

The datasets analyzed during the current study are available in the PubMed repository.

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
