# Peer review of "Ferroptosis, a Regulated Form of Cell Death, as a Target for the Development of Novel Drugs Preventing Ischemia/Reperfusion of Cardiac Injury, Cardiomyopathy and Stress-Induced Cardiac Injury"

_ijms, 2024, doi:10.3390/ijms25020897_

Round 1
Reviewer 1 Report
Comments and Suggestions for Authors
The review article titled 'Ferroptosis, a regulated form of cell death, as a potential therapeutic target for mitigating ischemia/reperfusion-induced cardiac injury, cardiomyopathy, and stress-related cardiac damage,' can aid readers in comprehending the mechanisms of ferroptosis and its involvement in the development of cardiac dysfunction, thereby serving as a foundation for drug development efforts. However, certain aspects require enhancement, such as...
Improvement Points:
- Visualize the Mechanisms: Provide graphical representations for both ferroptosis induction and its regulatory pathways. This will enhance the reader's understanding of the complex processes involved.
- Tailored Visualizations: For each cardiac injury (ischemia, cardiomyopathies, etc.), present specific graphical summaries illustrating how ferroptosis interacts and contributes to the pathology.
- Therapeutic Landscape: Discuss currently available or under-development natural/synthetic compounds targeting ferroptosis in each cardiac injury type. Emphasize their mechanisms of action and potential benefits.
- Charting the Future: Outline future directions for research based on current knowledge gaps and promising avenues for exploring ferroptosis-based therapies.
- Clinical Landscape (Optional): If available, mention specific compounds undergoing clinical trials for targeting ferroptosis in cardiac injury settings. Briefly summarize their potential and stage of development.
Overall:
By incorporating these suggestions, the authors can significantly enhance the review's clarity, impact, and translational potential, making it valuable for understanding and targeting ferroptosis in various cardiac pathologies.
Comments on the Quality of English LanguageMino modification is needed.
Author Response
Dear colleague. Thank you very much for your recommendation.
The review article titled 'Ferroptosis, a regulated form of cell death, as a potential therapeutic target for mitigating ischemia/reperfusion-induced cardiac injury, cardiomyopathy, and stress-related cardiac damage,' can aid readers in comprehending the mechanisms of ferroptosis and its involvement in the development of cardiac dysfunction, thereby serving as a foundation for drug development efforts. However, certain aspects require enhancement, such as...
Improvement Points:
Visualize the Mechanisms: Provide graphical representations for both ferroptosis induction and its regulatory pathways. This will enhance the reader's understanding of the complex processes involved.
We prepared Figure.
Tailored Visualizations: For each cardiac injury (ischemia, cardiomyopathies, etc.), present specific graphical summaries illustrating how ferroptosis interacts and contributes to the pathology.
We prepared Figure.
Therapeutic Landscape: Discuss currently available or under-development natural/synthetic compounds targeting ferroptosis in each cardiac injury type. Emphasize their mechanisms of action and potential benefits.
We added mini-chapter “Unresolved issues and prospects for the use of ferroptosis inhibitors for the treatment of cardiovascular diseases”
Charting the Future: Outline future directions for research based on current knowledge gaps and promising avenues for exploring ferroptosis-based therapies.
We added mini-chapter “Unresolved issues and prospects for the use of ferroptosis inhibitors for the treatment of cardiovascular diseases”
Clinical Landscape (Optional): If available, mention specific compounds undergoing clinical trials for targeting ferroptosis in cardiac injury settings. Briefly summarize their potential and stage of development.
I could not find any information on clinical trials of ferroptosis inhibitors in PubMed.
Overall:
By incorporating these suggestions, the authors can significantly enhance the review's clarity, impact, and translational potential, making it valuable for understanding and targeting ferroptosis in various cardiac pathologies.
Sincerely yours, Leonid N. Maslov
Reviewer 2 Report
Comments and Suggestions for Authors
The manuscript is extremely well written.
The authors are advised to highlight the drugs or antagonists of ferroptosis that might be useful for future drug discovery.
The role of non-coding RNA has been illustrated however the role of free circulating DNA/RNA is not detailed.
Ferroptosis and other cell death forms occur under very specific circumstances a paragraph can be used to mention the conditions.
A note to be included on the clinical onset of ferroptosis during cardiac pathophysiology and hypothesis towards clinical or pre-clinical detection of ferroptosis from blood parameters might be crucial for drug therapy.
Author Response
Dear colleague. Thank you very much for your recommendation.
The manuscript is extremely well written.
The authors are advised to highlight the drugs or antagonists of ferroptosis that might be useful for future drug discovery.
We added new mini-chapter “Unresolved issues and prospects for the use of ferroptosis inhibitors for the treatment of cardiovascular diseases”
The role of non-coding RNA has been illustrated however the role of free circulating DNA/RNA is not detailed.
We have cited all articles devoted to the cardioprotective or negative effects of non-coding RNAs. I added 4 more articles that I did not quote earlier. I apologize if I missed any article. Provide its PMID and we will quote this article.
Ferroptosis and other cell death forms occur under very specific circumstances a paragraph can be used to mention the conditions.
We added new Figure 1 on the hypothetical trigger mechanisms of ferroptosis.
A note to be included on the clinical onset of ferroptosis during cardiac pathophysiology and hypothesis towards clinical or pre-clinical detection of ferroptosis from blood parameters might be crucial for drug therapy.
An increase in the serum MDA, 4-HNE, 8-epi-prostaglandin F2α isoprostane levels demonstrated the presence of oxidative stress but is not indisputable evidence of the existence of ferroptosis. It is necessary to study the biopsy material for other markers of ferroptosis.
Sincerely yours, Leonid N. Maslov
Round 2
Reviewer 1 Report
Comments and Suggestions for Authors
The present manuscript is now quite impressive. The authors satisfactorily addressed each query.